# Predictors of Total Hip Arthroplasty Following Pediatric Surgical Treatment of Developmental Hip Dysplasia at 20-Year Follow-Up

**DOI:** 10.3390/jcm11237198

**Published:** 2022-12-03

**Authors:** Ernest Young, Christina Regan, Todd A. Milbrandt, Emmanouil Grigoriou, William J. Shaughnessy, Anthony A. Stans, A. Noelle Larson

**Affiliations:** Department of Orthopedic Surgery, Mayo Clinic, Rochester, MN 55905, USA

**Keywords:** developmental dysplasia of the hip, total hip arthroplasty, closed reduction, open reduction

## Abstract

Long-term outcomes of surgical treatment for pediatric developmental dysplasia of the hip (DDH) are not well defined. The purpose of this study was to report long-term radiographic and clinical outcomes, survivorship free of total hip arthroplasty (THA), and predictors of subsequent THA following childhood treatment of DDH. This study was a single-institution retrospective review of hips treated for DDH with closed or open reduction at a minimum 10-year follow-up. 107 patients (119 hips) were included with a mean patient age of 3.3 years at childhood treatment. At mean 30.5 years follow-up, 24 hips had undergone THA (20%). Mean patient age at time of THA was 33.5 years. None of the hips treated with closed reduction alone required THA, whereas 8 hips treated with open reduction (25%) underwent THA. Hips with patient age > 4 years at the time of treatment had lower survivorship at 35 years follow-up (50% vs. 85%; *p* < 0.001). Additionally, femoral osteotomy (OR 2.0, *p* < 0.001), and previous treatment elsewhere (27% vs. 16%; *p* < 0.01) were associated with subsequent THA. Early referral and appropriate intervention may prove important, as age and prior treatment were predictive of subsequent THA.

## 1. Introduction

Pediatric developmental hip dysplasia (DDH) describes a wide spectrum of pathology. A direct correlation between untreated DDH, subluxation or dislocation, and osteoarthritis of the hip has been well-described [1,2,3]. In infancy, bracing with a Pavlik harness remains the gold standard; however, dysplasia frequently presents at a later age in childhood, due to delay in diagnosis or treatment failure. Closed and open treatments have been well described and are used to achieve concentric reduction. Both in adjunct to these reduction techniques or as standalone procedures, femoral and pelvic osteotomies may be performed for residual dysplasia and to achieve better correction.

Despite widely accepted protocols for the treatment of DDH, there are very few long-term follow-up studies for this population. Longitudinal studies on the clinical results of open and closed reduction have shown that it can take many decades to determine the outcome of childhood intervention [4,5,6]. There are even fewer studies on pelvic osteotomies that follow patients into adulthood, [7] and thus many rely on surrogate or radiographic measurements in order to determine treatment indications [8,9,10].

The purpose of this study was to report long-term outcomes (>10 years following surgery) for the surgical treatment of DDH. Specifically, this study sought to report survivorship free of total hip arthroplasty (THA) and risk factors for subsequent THA.

## 2. Materials and Methods

This study was a retrospective review of patients with minimum 10-year follow-up. Institutional review board approval was obtained prior to initiation of the study (IRB 14-005476).

### 2.1. Study Design

All pediatric patients (<18 years old at the time of diagnosis) receiving treatment for a diagnosis of DDH from the years of 1976 to 2006 were identified with use of an institutional medical cross-referencing tool. This resulted in 1369 available charts for review. Records were reviewed including preoperative and postoperative course, radiographs and, when available, complications and orthopedic visits into adulthood pertaining to DDH. Exclusion criteria were (1) a pathologic/teratologic cause of hip dysplasia such as arthrogryposis, infection, spina bifida or other neuromuscular or other syndromic causes, (2) patients still under the age of 18 or deceased at the time of review, (3) patients treated with an abduction brace or a Pavlik harness alone, (4) adolescents (age > 10) treated with a PAO for residual hip dysplasia as their index surgical procedure, and less than 10 year follow-up available. Open reduction was performed in those patients who failed closed reduction and failed nonoperative treatment (abduction pillow or Pavlik harness) either at our center or a referring center. Surgeons at our center typically accepted a closed reduction if there is a very large safe zone, the hip is quite stable, and there is no medial dye pool. There were 225 patients (250 hips) treated surgically for DDH without associated conditions at our center during the study period. However, 118 patients (131 hips) had no follow-up > 10 years following their surgery and/or did not respond to our survey, and thus did not meet inclusion criteria. Therefore, 107 patients (119 hips) comprised our study cohort (Figure 1).

A survey was mailed to patients. Patients were asked to report any subsequent procedures on their dysplastic hip. Additional survey questions provided the validated outcome assessment for hip and health function, i.e., the Oxford Hip Score, the modified Harris Hip Score, and the EQ-5D [11,12,13].

Radiographic analysis was performed on preoperative and postoperative images when available for review. Hip radiographs in skeletally mature patients were reviewed for evidence of osteoarthritis, acetabular angle, and center-edge angle (CEA) of Wiberg [14,15]. Arthritis was assessed using the Kellgren-Lawrence classification system [16]. The CEA was assessed using the Severin Classification [17,18]. In this system a normal CEA is defined as greater than 20 in children under 14 and greater than 25 in children 14 and over. Class I and Class II are defined as having normal CEA but Class II has moderate deformity of the femoral head, neck or acetabulum. Class I and II are further subdivided into subclasses A and B depending on the amount of CEA (A is normal CEA of greater than 20 and 25 in the appropriate ages and B is a CEA of 16–19 in ages under 14 and 20–25 in ages 14 and above). Class III was defined as dysplasia without subluxation; CEA less than 15 in ages under 14 and less than 20 in ages 14 and above. Class IV was subluxation classified into two groups (IVa having a CEA greater than 0 and IVb having CEA less than 0).

Patients were stratified by age and treatment type. Pelvic osteotomies were ranked by increasing scale of intervention. A pelvic osteotomy alone was ranked low intervention, combined femoral and pelvic osteotomy was high, and combined open reduction and pelvic osteotomy was ranked as intermediate for level of intervention. Salter et al. previously reported improved clinical outcomes following pelvic osteotomy in children ages 1.5 to 4 compared to those >4 years of age [19]. Thus, patients were grouped by age at the time of surgery, >4 and ≤age 4. Comparisons between groups were made for major outcome measures, including clinical outcome, radiographic arthritis, and survivorship free of THA.

### 2.2. Statistics

For comparison of independent measures, Wilcoxon Rank-Sum test was used for continuous variables and Chi-Square analysis on categorical. Continuous variables under review failed to provide normative data, and non-parametric evaluation was thus used. A *p* value less than 0.05 was considered significant. Survival analysis censored patients by interval between surgery and last follow-up or by interval to subsequent THA. Multivariable analysis of risk factors for subsequent THA was performed using logistic regression and Cox proportional-hazards technique. For patients with surgical management of bilateral hips, each hip was considered independent and separated into two data entries.

## 3. Results

### 3.1. Patient Demographics

A total of 107 patients (119 hips) met inclusion criteria. The average age of the patient at index procedure was 3.3 (range 0.1 to 18) years old (Table 1). The average length of follow up, from surgery to evaluation, was 30.5 years (with a range of 10 to 46). Actual patient age at latest follow up was a mean 35 years (range 20 to 47). There were 12 males and 95 females, with 1 male and 11 females with bilateral surgery in childhood.

The two largest categories of treatment were closed reduction (28 patients, 24%) and the pelvic osteotomy group (35 patients, 29%). A femoral osteotomy was performed in isolation for 10 hips (8%). Furthermore, there were 15 hips (13%) treated with open reduction, and 17 hips (14%) treated with open reduction and pelvic osteotomies (Table 1).

Of the 66 pelvic osteotomies, 30 had prior attempt at closed reduction and 15 had a prior open reduction. Osteotomy types included: Salter (55), Pemberton (8), Sunderland modifications of the Salter (2), and a triple osteotomy of Steel (1) [19,20,21,22]. For those treated previously, the prior treatment was performed at our institution for 24 hips and at an outside facility for 21 hips. A revision osteotomy was performed for 4 patients, all of which had a prior pelvic (Salter type) osteotomy performed elsewhere.

Additional subsequent procedures following initial surgical treatment included a femoral varus derotational osteotomies (11 hips) and a Bernese-type periacetabular osteotomy (4 hips). All of these 15 cases had an index treatment with a pelvic osteotomy at our institution. Of the hips originally treated with closed reduction, 3 had subsequent procedures: hip arthroscopy (2 hips), and a triple osteotomy of Steel (1 hip). All femoral procedures were classified as femoral osteotomies.

### 3.2. Radiographic Assessment

Postoperative radiographs were available for acetabular angle, CEA and Severin classification for 46 hips (Table 2). There were no known radiographic or clinically reported cases of acetabular osteonecrosis. However, 3 hips, 2 femoral osteotomies and 1 combined femoral and pelvic osteotomy, showed radiographic evidence of femoral avascular necrosis.

Follow-up radiographs taken at least 10 years after surgery were available for 23 hips that had not gone on to THA. High grade osteoarthritis (KL grade 3 or 4) was present for 13 hips (Table 3). The remainder showed only mild osteoarthritis (example, Figure 2).

### 3.3. Total Hip Arthroplasty

Of the 119 hips, a total of 24 patients (20%) went on to have total hip arthroplasty at a mean 27.0 years following surgery for DDH (range 13 to 42). Mean patient age at THA was 33.5 years (range 20 to 42). Of note, none of the hips that had a closed reduction went on to THA, whereas four of the hips treated with open reduction alone (27%) and four of the hips treated with open reduction and pelvis osteotomy (24%) went on to THA. (Figure 3).

The 10 hips treated in childhood at our institution with a femoral osteotomy alone represent a unique subset of patients with severe disease and resultant poor outcomes. Four of the hips had already had prior treatment at an outside institution and 3 of these hips had prior Salter osteotomy. They had a treatment at an older mean age (7.3 years with range 1.5 to 19.6) and 5 of these hips (50%) went on total hip arthroplasty at a mean age of 27.5 years (range 20–33). Femoral osteotomy alone was a significant risk factor for conversion to THA on logistic regression analysis (OR 4.9, CI 1.2–18.0, *p* = 0.02).

Of the hips that underwent a pelvic osteotomy alone, only 3 of 35 (8.5%) had a subsequent THA. In contrast, 4 of 17 (23.5%) of hips that received an open reduction and pelvic osteotomy and 8 of 14 (57%) of hips that had combined pelvic and femoral osteotomies had a subsequent THA (*p* < 0.0001). Subsequent THA was not dependent on the type of pelvic osteotomy (Salter vs. not), side, gender, prior treatment or a prior closed or open reduction (*p* > 0.05). Patients who had a femoral osteotomy at the same time or following a pelvic osteotomy were at higher risk of undergoing THA in adulthood (OR 9.5, CI 3.5–26.1, *p* < 0.0001).

Logistic regression analysis for subsequent THA (excluding the femoral osteotomy only hips) showed an association with age at time of treatment and increasing scale of pelvic osteotomy intervention as independent predictors (*p* < 0.05). Additionally, the 3 cases of femoral avascular necrosis went on to total hip arthroplasty at 26, 26, and 27 years of age following index treatment.

Survivorship free of THA for hips that underwent a pelvic osteotomy was evaluated. Overall survivorship from of THA for the 66 hips who underwent childhood pelvic osteotomy was 95% at 25 years, however, declined steeply to 70% at 35 years following surgery (Figure 4). Further analyses showed that pelvic osteotomy alone had a 90% survivorship at 35 years while combined open reduction and pelvic osteotomy showed 80% at 25 years and 60% survivorship at 35 years. Moreover, combined femoral and pelvic osteotomies showed 90% at 25 years, but only 50% survivorship at 35 years.

When patients were stratified by age, there was a significant difference in survivorship. Patients more than 4 years of age were compared to those 4 years and younger at the time of a pelvic osteotomy (Figure 5). Those greater than age 4 at time of surgery had higher rate subsequent THA (OR 4.8, CI 1.8–12.4, *p* = 0.0013). Survivorship for hips 4 years or younger was 95% at 25 years and 85% for those older 4. Furthermore, survivorship free from THA at 35 years was 85% for hips age 4 and under, while those older than 4 showed only 50% survivorship (*p* < 0.01).

### 3.4. Clinical Outcomes

Survey data was available for 76 patients (84 hips). Of the 76 patients (84 hips) that completed and returned the survey, 64 patients (70 hips) had not received a THA (Table 4). The Harris Hip Score (HHS) was scored out of a maximum of 96, the Oxford Hip Score (OHS) was scored out of a maximum of 48 and the EQ5D was scored out of a maximum of 1.0. Overall, outcomes for those treated with closed reduction and open reduction with or without pelvic osteotomy were good. No differences between these treatment groups were identified for each of the subjective outcome measures. However, the femoral osteotomy alone as well as those treated with combined femoral and pelvic osteotomies showed significantly lower outcomes (*p* < 0.05) for the HHS, OHS, and EQ5D (EQ5D was not significantly lower for combined femoral and pelvic osteotomies, *p* > 0.05).

## 4. Discussion

Treatment of DDH in children older than 6 months of age remains a challenge in today’s practice. The goal is to achieve concentric hip joint as early as possible without developing complications such as avascular necrosis [5]. The capacity of the acetabulum to remodel is dependent on the age at which the hip is reduced [23,24]. Thus, closed and open reductions are well-established treatments for dysplasia for children younger than 1.5 years and remain standard of care for a dislocated hip. However, the indications for pelvic osteotomy are less clear, with limited data available in the literature. Timing and the concomitant use of open reduction or femoral osteotomy are largely institution dependent. Despite this controversy, pelvic osteotomies are commonly performed in an attempt to improve long-term function for the dysplastic hip. Osteoarthritis, pain, dysfunction as well as subsequent THA at a young age are all common sequelae of DDH. As shown in previous long-term studies, hip osteoarthritis and progression to THA requires decades to become symptomatic, and may not appear until the third or fourth decade of life [4,5,6,7]. DDH patients presenting for THA have their share of surgical challenges stemming from various bony morphologies [24,25,26,27]. Few mid to long-term clinical follow up studies exist, especially for those receiving pelvic osteotomies. The purpose of this study was to provide long-term follow up of patients who underwent surgical treatment for DDH. Age at the time of pelvic osteotomy as well as any adjunct treatments, such as femoral osteotomy, were independent predictors of subsequent THA, which likely are markers of disease severity at presentation.

There are only three long-term studies on the Salter osteotomy in the literature. Bohm et al. reported outcomes at a mean 30 years after pelvic osteotomy, Thomas et al. reported Dr. Salter’s original cohort at mean 43 years following surgery, and Scott et al. reported outcomes at a mean of 45 years after either closed reduction or open reduction with Salter innominate osteotomy [7,28,29]. All studies showed between 77 and 100% survivorship at 30 and 40 years. Similarly, patients in our cohort who had early osteotomy, under the age four, and or had pelvic osteotomy alone showed good survivorship free of subsequent THA. Survivorship at 35 years was lower, roughly 80% in our cohort, but is consistent with the results reported by Thomas et al. with 54% survivorship at 45 years following surgery and those reported by Scott et al. with 55% survivorship of closed reduction patients and 63% of open reduction/osteotomy patients [7,29].

However, results were remarkably worse for patients with surgery over the age four, combined femoral osteotomies, or combined open reduction and pelvic osteotomy. Thomas et al. presented a cohort with a maximum age of 4.7 years, mean 2.8 years, at the time of surgery, and their cohort was not treated with subsequent or combined femoral osteotomies. Again, survivorship was only 54% at 45 years. Similarly, Scott et al. found worse outcomes in patients that had required bilateral reductions done at or after 3 years of age. In our cohort, the patients who had surgery over the age of 4 had a 50% survival at 35 years and those who had combined femoral and pelvic osteotomies had only 45% survival at 35 years. These results are also consistent with long-term outcomes of patients undergoing open reduction for DDH [5,29]. Malvitz et al. found a marked increase in osteoarthritis and a decrease in outcomes between in the third- and fourth-decades of life following surgery [5].

Timing of reduction and osteotomies remain dogmatic, based on evidence that supports a limited window for acetabular remodeling [30]. Our study does support current practice, in that the timing of surgical intervention had a significant impact on long-term outcome. Several patients received a Salter osteotomy over the age of 8 before the advent of the Bernese periacetabular osteotomy. Many of these late presenting hips received a femoral varus derotational or shortening osteotomy as their only definitive treatment, and a subsequent THA was performed on 50% at only 28 years. Although femoral osteotomies have been utilized for many decades, its use is usually in adjunct to pelvic osteotomy and may be a marker for severe dysplasia in this cohort of patients with resultant poor outcomes and high rate of THA in adulthood.

Closed reduction and open reduction showed 100% and 90% survivorship, respectively, free of THA at 35 years follow surgery. This is consistent with other large series [4,5,6]. Additionally, this subpopulation had superior patient reported outcomes at latest follow-up. Of note, our center has exacting standards for accepting a closed reduction and will turn to open reduction in a very young patient even if the closed reduction is not perfect on arthrogram.

There are numerous limitations of this study including lack of standardized treatment protocol and the retrospective nature. This cohort of patients encompassed over 40 years of treatment for DDH at a busy tertiary care practice. Patients with more severe radiographic disease were treated with a greater level of intervention. Follow-up bias that may have increased reported rates of secondary surgery and decreased outcome scores, as patients with residual dysplasia or symptomatology may be more likely to continue follow-up past skeletal maturity. Thus, there is an inherent selection bias in this cohort of patients, with patients with underlying significant dysplasia, later age and presentation, and previous failed surgery treated with more osteotomies and combined procedures. Lastly, radiographic follow up was limited, and failed to provide significant insights. A significant number of patients were lost-to-follow-up and did not meet inclusion criteria (118 patients, 131 hips). As with any survey study, there may be an element of reporter bias which is inherent in U.S. survey studies which follow pediatric patients into adulthood [31]. Despite limitations, there were several strengths of this study. Long-term follow-up for DDH was provided on 119 hips that underwent surgical treatment in childhood. This is a large cohort of patients in comparison to the few cohorts reported to date. Additionally, outcomes were comprehensive, using patient-reported outcomes, radiographic review, and survival analysis for subsequent THA. Lastly, long-term outcomes of femoral osteotomy for DDH have not been clearly reported. This study showed inferior outcomes for femoral osteotomy, likely as a marker for more significant dysplasia. Modern powerful correction strategies such as periacetabular osteotomy may eliminate the need for femoral osteotomy in older children.

Among the hips treated with pelvic osteotomy, there were higher rates of THA with increasing level of intervention (pelvic osteotomy alone vs. combined open reduction and pelvic osteotomy vs. combined femoral and pelvic osteotomy). Of the hips that underwent a pelvic osteotomy alone, only 3 of 35 (8.5%) had a subsequent THA. In contrast, 4 of 17 (23.5%) of hips treated with open reduction and pelvic osteotomy and 8 of 14 (57%) of hips that had combined pelvic and femoral osteotomies had a subsequent THA (*p* < 0.05). Thus, increased age and disease severity likely prompted higher levels of intervention which failed to return normal function to these hips. Outcomes were poor compared to hips treated successfully at a young age with closed reduction or open reduction.

In summary, this study reports mean 30-year outcomes of surgical treatment for pediatric developmental hip dysplasia. Few studies have reported long-term outcome of pediatric patients treated with pelvic and femoral osteotomies for DDH. Our cohort demonstrated an 8.5% rate of THA in patients who underwent pelvic osteotomy alone, 23.5% rate of THA for pelvic osteotomy and open reduction, and 57% rate of THA for patients with combined pelvic and femoral osteotomies. Increased rate of THA is associated with underlying disease severity reflected by older age at treatment and need for revision surgery. At mean 36.7 year follow-up, 28 patients treated successfully with closed reduction alone had excellent results whereas ¼ of patients treated with open reduction with or without pelvic osteotomy required THA. This data can be used to counsel families of children undergoing DDH treatment. Age at time of surgery as well as combined procedures were associated with worse survival in the long term, likely representing worse dysplasia and deformity prior to childhood treatment.

## Figures and Tables

**Figure 1 jcm-11-07198-f001:**
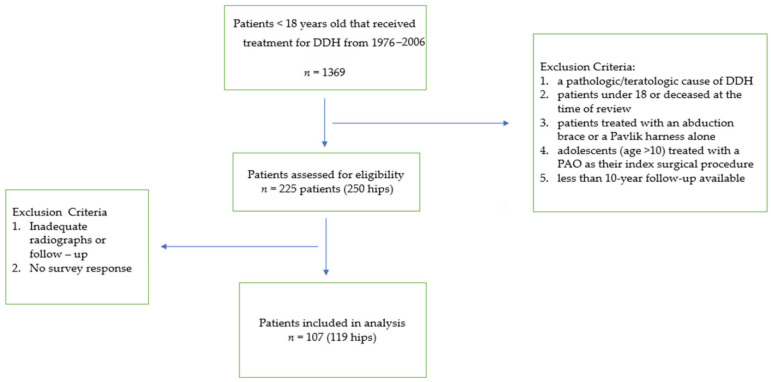
Patient selection process and final cohort after exclusion criteria applied.

**Figure 2 jcm-11-07198-f002:**
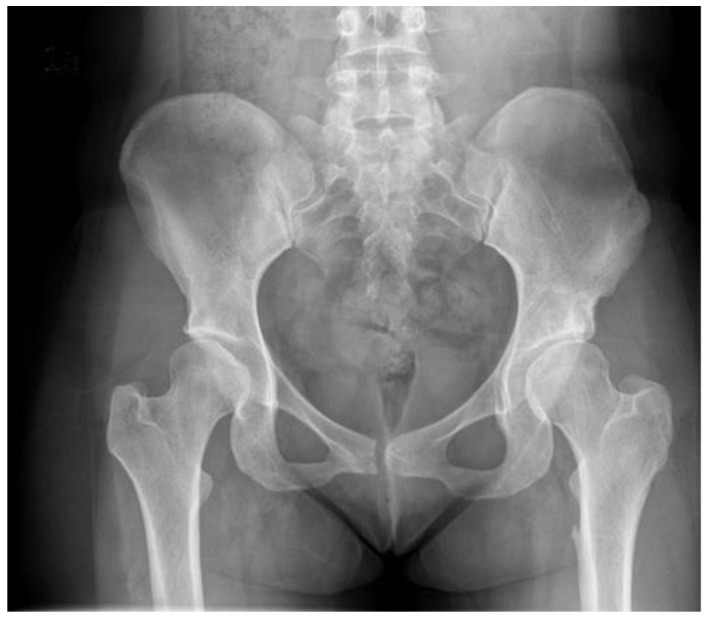
AP pelvis in patient 20 years following open reduction and pelvic osteotomy shows mild Grade 2 bilateral osteoarthritis.

**Figure 3 jcm-11-07198-f003:**
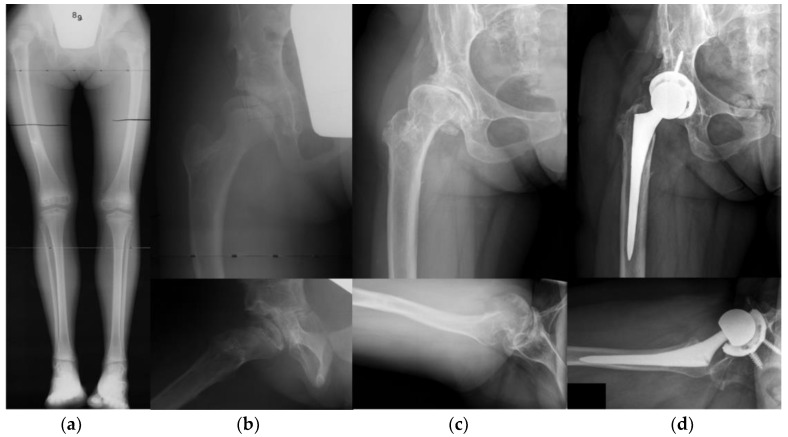
Case of an 8 year-old female treated with Salter osteotomy and femoral shortening who went on to subsequent THA at age 37. (**a**) Presenting long-leg standing radiograph. (**b**) AP and frog-leg radiographs following Salter osteotomy and femoral shortening. (**c**) Patient at age 37, 29 years following combined pelvic and femoral osteotomies, with end-stage osteoarthritis. (**d**) AP and cross table radiographs following THA.

**Figure 4 jcm-11-07198-f004:**
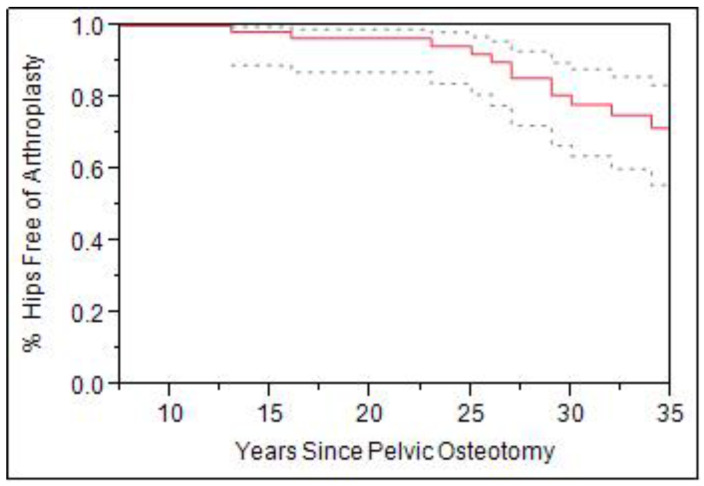
Survivorship curve (Solid line) with 95% confidence intervals (dashed line) for all pelvic osteotomies with total hip arthroplasty as the end point (*n* = 66).

**Figure 5 jcm-11-07198-f005:**
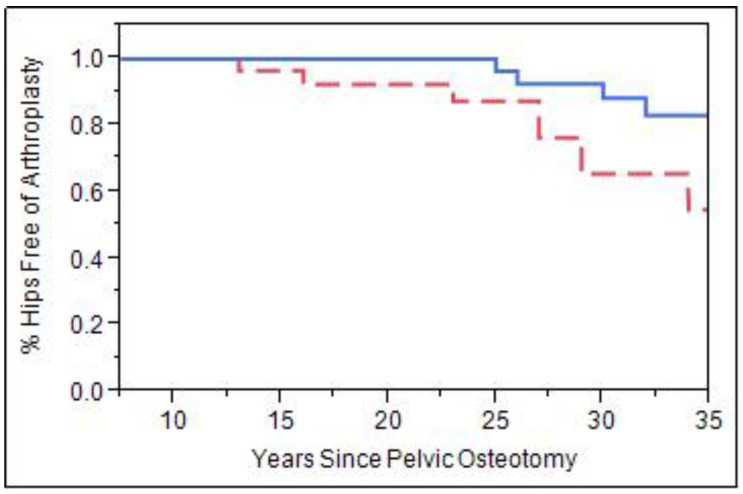
Survivorship curve examining patients that received a pelvic osteotomy at age 4 and under at time of surgery (solid blue line, *n* = 34) vs. patients that had pelvic osteotomy at age 4 or older at time of surgery (dashed red line, *n* = 18, *p* < 0.01).

**Table 1 jcm-11-07198-t001:** Distribution of treatment and ages for the cohort.

Type of Treatment	Number of Hips	Age at Treatment	Years of Follow-Up	Total Hip Arthroplasty at Latest Follow-Up
Closed Reduction	28	1.1 ± 1.5	36.7 ± 7.1	0 (0%)
Open Reduction	15	0.6 ± 0.4	31.6 ± 9.3	4 (27%)
Open Reduction and Pelvic Osteotomy	17 (15 Salter 1 Steele 1 Pemberton)	2.7 ± 2.7	26.1 ± 8.9	4 (24%)
Pelvic Osteotomy	35 (27 Salter 1 Steele 7 Pemberton)	4.5 ± 3.7	27.4 ± 10.0	3 (8.5%)
Femoral Osteotomy	10	7.0 ± 6.7	28.5 ± 5.0	5 (50%)
Combined Pelvic and Femoral Osteotomy	14 (13 Salter 1 Pemberton)	6.2 ± 2.3	32.0 ± 6.9	8 (57%)

**Table 2 jcm-11-07198-t002:** Analysis of postoperative radiographs per treatment type.

Type of Treatment	Acetabular Index	Center Edge Angle	Severin Score
Closed Reduction (*n* = 7)	27.1 ± 10.4	21.2 ± 6.0	Ia—4 IIb—2 III—1
Open Reduction (*n* = 5)	39.3 ± 6.4	25.9 ± 13.8	Ia—2 IIa—1 III—1 IVa—1
Pelvic Osteotomy (*n* = 21)	37.6 ± 13.4	27.8 ± 12.4	IIa—8 III—7 IVa—1
Open Reduction/Pelvic Osteotomy (*n* = 11)	43.5 ± 6.0	22.4 ± 11.4	IIa—4 III—5 IVa—1 IVb—1
Femoral Osteotomy (*n* = 3)	38.5 ± 14.8	20 ± 13.0	III—2 IVa—1
Combined Pelvic/Femoral Osteotomy (*n* = 4)	39.5 ± 4.0	17.7 ± 19.8	III—2 IVa—2
All Hips (*n* = 46)	37.8 ± 11.5	24.0 ± 12.9	Ia—6 IIa—13 IIb—2 III—18 IVa—6 IVb—1

**Table 3 jcm-11-07198-t003:** Analysis of Kellgren-Lawrence grade of osteoarthritis per treatment type at minimum 10 Years Following Surgery.

Type of Treatment	KL Grade	*n*	Mean Time from Surgery (Years)
Closed Reduction (*n* = 4)	Grade 1	-	32.2 ± 11.6
	2	3	
	3	1	
	4	-	
Open Reduction (*n* = 1)	Grade 1	1	38.4 ± 0
	2	-	
	3	-	
	4	-	
Pelvic Osteotomy (*n* = 6)	Grade 1	-	31.5 ± 10.3
	2	3	
	3	-	
	4	3	
Open Reduction/Pelvic Osteotomy (*n* = 7)	Grade 1	-	17.4 ± 5.2
	2	1	
	3	4	
	4	2	
Femoral Osteotomy (*n* = 2)	Grade 1	2	9.3 ± 0
	2	-	
	3	-	
	4	-	
Combined Pelvic/Femoral Osteotomy (*n* = 3)	Grade 1	-	29.3 ± 9.5
	2	-	
	3	2	
	4	1	
All Hips (*n* = 23)	Grade 1	3	24.9 ± 12.0
	2	7	
	3	7	
	4	6	

**Table 4 jcm-11-07198-t004:** Clinical outcomes per treatment type.

Type of Treatment	Harris Hip Score	Oxford Hip Score	EuroQual5D
Closed Reduction	89.3 ± 11.5	42.5 ± 8.9	0.94 ± 0.1
Open Reduction	83.9 ± 18.8	43.0 ± 7.9	0.93 ± 0.08
Pelvic Osteotomy	87.2 ± 9.1	41.1 ± 6.7	0.88 ± 0.12
Open Reduction and Pelvic Osteotomy	92 ± 3.5	43.7 ± 5.0	0.92 ± 0.07
Femoral Osteotomy	69.8 ± 22.4 *	29.3 ± 14.8 *	0.80 ± 0.06 *
Combined Pelvic and Femoral Osteotomy	71.4 *	37.3 ± 5.7 *	0.87 ± 0.11

* *p* < 0.05.

## Data Availability

Not applicable.

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
