# Peer review of "Predictors of Total Hip Arthroplasty Following Pediatric Surgical Treatment of Developmental Hip Dysplasia at 20-Year Follow-Up"

_jcm, 2022, doi:10.3390/jcm11237198_

Round 1
Reviewer 1 Report
This paper presents a retrospective review of outcomes following pediatric treatment of DDH. While there are obviously limitations with this type of study, this is a very important problem that is difficult to study, and as the authors' correctly point out, limited long term follow-up data exists. This represents an admirable work in adding to this literature.
General comments
This paper does a good job of validating some of the current dogma in treating DDH, namely that age of treatment and severity of disease are likely to be important in predicting outcomes, and produce some important numbers in terms of survivorship and clinical outcomes, that are likely to be very helpful in counseling patients and their families in the future.
Specific comments:
Table 1: it is surprising to me that patient's receiving open reduction alone are typically younger than those receiving closed reduction alone. Do the authors have an explanation for this? An explanation of their typical institutional protocol in terms of treatment of DDH would be helpful. I'm sure this has changed over time and varies by surgeon, but a general outline would still be helpful.
Lines 104-106: authors state that 76 patients (84 hips) had survey data available and 31 patients (35 hips) had clinical/radiographic follow-up available. I believe what they mean is that 76 patients (84 hips) had both survey data and clinical/radiographic follow-up available and 31 patients (35 hips) had only clinical/radiographic follow-up available. If so, the language could be updated to make this more clear.
Lines 112-115: this paragraph flips back and forth between presenting absolute numbers and percentages. I believe it would be more clear if consistent (whether absolute numbers, percentages, or both).
Lines 132-136: it's unclear to me if these radiographic parameters are at last follow-up or immediately postoperative. Clarification of time frame would be helpful.
The results of the univariate and multivariate logistic regression are a bit hidden for me. I would appreciate a table showing clearly the included covariates with their univariate and multivariate OR and 95% CIs.
Author Response
Table 1: it is surprising to me that patient's receiving open reduction alone are typically younger than those receiving closed reduction alone. Do the authors have an explanation for this? An explanation of their typical institutional protocol in terms of treatment of DDH would be helpful. I'm sure this has changed over time and varies by surgeon, but a general outline would still be helpful.
Thank you so much for the suggestion. We have adjusted the methods section to reflect your recommendations. We have added in Methods, “Open reduction was performed in those patients who failed closed reduction and failed nonoperative treatment (abduction pillow or Pavlik harness) either at our center or a referring center. Surgeons at our center typically accepted a closed reduction if there is a very large safe zone, the hip is quite stable, and there is no medial dye pool.” Also, in discussion, “Of note, our center has exacting standards for accepting a closed reduction and will turn to open reduction in a very young patient even if the closed reduction is not perfect on arthrogram.”
Lines 104-106: authors state that 76 patients (84 hips) had survey data available and 31 patients (35 hips) had clinical/radiographic follow-up available. I believe what they mean is that 76 patients (84 hips) had both survey data and clinical/radiographic follow-up available and 31 patients (35 hips) had only clinical/radiographic follow-up available. If so, the language could be updated to make this more clear.
Thank you so much for this edit. We agree that the wording could be better and have revised the paper to reflect your changes. The manuscript now reads, “A total of 107 patients (119 hips) met inclusion criteria. Survey data was available for 76 patients (84 hips). Of the 76 patients (84 hips), 31 patients (35 hips) had clinical or radiographic follow-up >10 years from their surgery.”
Lines 112-115: this paragraph flips back and forth between presenting absolute numbers and percentages. I believe it would be more clear if consistent (whether absolute numbers, percentages, or both).
Thank you so much for this edit. We agree that the wording could be better and have revised the paper to reflect your changes. The manuscript now reads, “The two largest categories of treatment were closed reduction (28 patients, 24%) and the pelvic osteotomy group (35 patients, 29%). A femoral osteotomy was performed in isolation for 10 hips (8%). Furthermore, there were 15 hips (13%) treated with open reduction, and 17 hips (14%) treated with open reduction and pelvic osteotomies”
Lines 132-136: it's unclear to me if these radiographic parameters are at last follow-up or immediately postoperative. Clarification of time frame would be helpful.
Thank you so much for your revision. We agreed that this needs to be clarified. In the initial paragraph (lines 132-136) we talk about radiographs done immediately post-op. In the subsequent paragraph we talk about the radiographs done at most recent follow up. That second paragraph was placed after Table 2. We reformatted that second paragraph so that it directly follows the first, improving the flow of the paper. Thank you so much for your recommendation.
The results of the univariate and multivariate logistic regression are a bit hidden for me. I would appreciate a table showing clearly the included covariates with their univariate and multivariate OR and 95% CIs.
We have used logistic regression for odds ratios, and have added confidence intervals to the text. I don’t think there is enough here to justify a formal table. Please see changes in the text.
Reviewer 2 Report
Thank you for the opportunity to review. I have some specific remarks as following.
1. It is unclear what kind of cases are targeted in this study. Are cases with less than 10 years of imaging follow-up not excluded?
2. Please specify the number of each case for exclusion criteria. How many cases are excluded with less than 10 years of follow-up? This information is necessary to consider selection bias.
3. The purpose of this research is ambiguous. There are a variety of childhood hip surgical treatments. However, simple comparison is difficult because each operation has different indications. Moreover, the small number of cases also makes evaluation difficult.
4. Survival curves and multivariate analysis results are only partially described. It is unclear what selection criteria are used for analysis.
Author Response
- It is unclear what kind of cases are targeted in this study. Are cases with less than 10 years of imaging follow-up not excluded?
Thank you so much for this comment. We included patients with and without imaging at final follow-up. Patient either had responded to a survey at final follow-up and/or had a clinical visit at final follow-up. Our aim was to include patients with long term follow up after their initial DDH surgery which is extremely challenging in the US. We excluded patients that had less than 10 years of follow up after their initial procedure. This provided us with a range that spanned from 10 years of follow up to 46 years of follow up with a mean of 30.5 years. Patients were excluded from final analysis if they did not respond to the survey or did not have clinical follow-up (visits or x-rays).
- Please specify the number of each case for exclusion criteria. How many cases are excluded with less than 10 years of follow-up? This information is necessary to consider selection bias.
Thank you so much for your revision. In lines 61 to 63, we talk about how 118 patients (131 hips) were excluded because they either did not have adequate radiographic evidence, did not have clinical follow up > 10 years after initial intervention, did not respond to our survey, or did not meet inclusion criteria. There certainly is selection bias, but this is a mean 30-year follow-up study, so 80% follow-up at 30 years cannot be expected and there is little published on this topic. We have added these numbers to the limitations section.
- The purpose of this research is ambiguous. There are a variety of childhood hip surgical treatments. However, simple comparison is difficult because each operation has different indications. Moreover, the small number of cases also makes evaluation difficult.
Thank you so much for your feedback. This is a mean 30-year follow-up study of patients treated operatively in childhood for hip dysplasia and our primary outcome measure is need for total hip arthroplasty. There is very limited data available on this topic. We think this is an important contribution to the medical literature. We have a total of 107 patients and 119 hips. This is actually not that small of a series.
- Survival curves and multivariate analysis results are only partially described. It is unclear what selection criteria are used for analysis.
Thank you so much for your revision. Our overall analysis was conducted on the 107 patients (119 hips) that met our inclusion criteria. From this base population we were able to conduct several analyses. We used Kaplan Meier curves for Figure 4 and Figure 5 to predict survival free from total hip arthroplasty. Figure 4 has a survivorship curve free from total hip arthroplasty for the 66 patients treated with pelvic osteotomies. Figure 5 included patients treated with a pelvic osteotomy at age 4 and under at time of surgery (solid blue line, n=34) vs patients that had pelvic osteotomy at age older than 4 at time of surgery (dashed red line, n=18, p < 0.01).
With the 46 hips that had postoperative radiographic images, we conducted an analysis of the acetabular angle, CEA, and Severin classification. With the 23 hips that had 10+ year follow up imaging, we were able to analyze the Kellgren-Lawrence grade and mean time to the next surgery. We have added more details to the statistical analysis.
We have clarified KM survival results. “Survivorship free of THA for hips that underwent a pelvic osteotomy was evaluated. Overall survivorship from of THA for the 66 hips who underwent childhood pelvic osteotomy was 95% at 25 years, however, declined steeply to 70% at 35 years following surgery (Figure 4). Further analyses showed that pelvic osteotomy alone had a 90% survivorship at 35 years while combined open reduction and pelvic osteotomy showed 80% at 25 years and 60% survivorship at 35 years. Moreover, combined femoral and pelvic osteotomies showed 90% at 25 years, but only 50% survivorship at 35 years.”
